# mGEV: Extension of the GEV Activation to Multiclass Classification

**Joshua Bridge**[1]                                                                    JBRIDGE@LIVERPOOL.AC.UK

**Yalin Zheng**[1]                                                                      YZHENG@LIVERPOOL.AC.UK

[1] *Department of Eye and Vision Science, Institute of Life Course and Medical Sciences, University of Liverpool, Liverpool, UK*

## Abstract

Unbalanced data poses a challenge when training machine learning algorithms; the algorithm often overfits on the dominant class and neglects the smaller classes. While methods such as oversampling aim to rebalance the data, this can lead to overfitting. When a certain class is underrepresented, either because it is a rare disease or few images exist, methods are needed to adequately account for this. The generalized extreme value (GEV) activation has recently been proposed as a solution to highly unbalanced data; however, the GEV activation is only available for binary classification. We extend this to the multiclass case with the multiclass GEV (mGEV) activation. We conduct experiments on X-ray images, with three classes, showing much-improved performance over the commonly used softmax activation. Code for the mGEV activation is available at [https://github.com/JTBridge/GEV].

**Keywords:** Classification, Deep Learning, Activation function

## 1. Introduction

When developing multiclass classification algorithms, the datasets used are often highly unbalanced, with one class far outweighing others. Methods available to overcome this problem often rely on oversampling the dataset or weighting the loss function (Menon et al., 2021; Cui et al., 2019). The generalized extreme value (GEV) activation (Bridge et al., 2020) was proposed as an alternative. When the data is balanced, the GEV becomes approximately equivalent to the sigmoid activation. However, trainable parameters within the GEV allow it to change shape and better model the longtailed distribution. The most significant limitation of this approach is that it is only available on binary classification problems, limiting its applicability. We aim to extend the GEV activation to the multiclass case and propose the multiclass GEV (mGEV) activation function in this work.

## 2. Methods

The backbone classification network in this work consists of InceptionV3 (Szegedy et al., 2016), followed by a pooling layer and a dense layer with an activation function. For the activation function, we compare the proposed mGEV activation with the commonly used softmax activation. The original GEV activation function (Bridge et al., 2020) is given by

$$GEV(x|\mu,\sigma,\xi) = \begin{cases} exp\left\{-exp\left(-\frac{x-\mu}{\sigma}\right)\right\}, & \text{if } \xi = 0, \\ exp\left\{-\left\{1+\xi\left(\frac{x-\mu}{\sigma}\right)\right\}^{-1/\xi}\right\}, & \text{if } \xi \neq 0, \end{cases}$$

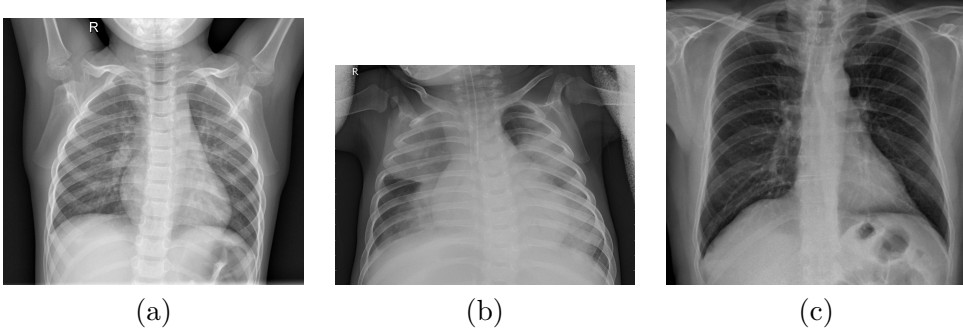

(a) (b) (c)

Figure 1: Examples images of (a) normal, (b) non-COVID pneumonia, and (c) COVID-19.

where $\mu$, $\sigma$, and $\xi$ are parameters to be learned in the deep learning framework. According to the extreme value theorem, the properly normalized maximum of a sample of independent and identically distributed random variables can only converge to the GEV distribution. The GEV distribution is often used in finance to model tail risks. Like the sigmoid activation, the GEV activation rescales the values between zero and one to give a probability. However, the parameters allow the curve to better model the longtailed distribution that occurs with extreme data. To extend the GEV to the multiclass problem, we propose fitting a curve for each class that produces a probability for each class. The probabilities are then normalized using

$$mGEV(GEV) = \frac{GEV}{\Sigma_i GEV}.$$

This only increases the number of parameters by $2 \times N_{classes} + 1$ over the softmax activation function. Like the softmax activation, the mGEV activation can potentially suffer from vanishing gradients; we mitigate this risk by applying L1 and L2 regularization to the parameter $\xi$ of $1e-5$ each and clipping the input from the full-connected layer to between -20 and 20.

## 3. Experiments

We demonstrate the proposed mGEV activation on a publicly available dataset the same data as used in the COVID vs. other experiments in the original GEV paper (Bridge et al., 2020), consisting of normal and non-COVID pneumonia (Kermany et al., 2018) with COVID-19 X-rays added (Cohen et al., 2020). We split the normal and pneumonia classes to create a three-class problem rather than binary. Example images are shown in Figure 1. For testing we used, 234 normal, 390 pneumonia, and 100 COVID-19 X-rays; this left 1349, 3883, and 29 images for training; we used 30% of these for validation.

Results show that the mGEV activation gives improved overall performance over the commonly used softmax activation, with an increase in recall for the under-represented classes and only a minor decrease for the over-represented class. Increases in macro averages for precision, recall, and F1-Score show that overall the method improves performance, giving more balanced results. Full results are shown in Table 1.

Table 1: Results on the hold-out testing set showing the recall, precision, and F1-Scores.

|  |  | Normal | Pneumonia | COVID-19 | Macro Average |
|---|---|---|---|---|---|
| Softmax | Recall | 0.54 | 1.00 | 0.77 | 0.77 |
|  | Precision | 0.99 | 0.75 | 1.00 | 0.91 |
|  | F1-Score | 0.70 | 0.86 | 0.87 | 0.81 |
| mGEV | Recall | 0.68 | 0.99 | 0.82 | 0.83 |
|  | Precision | 0.97 | 0.80 | 1.00 | 0.92 |
|  | F1-Score | 0.80 | 0.89 | 0.90 | 0.86 |

## 4. Conclusions

Here, we have extended the GEV activation to the multiclass case. Overall model performance was improved, with an improvement in recall for the under-represented classes. As this method replaces the activation function only, it can be combined with other methods which rebalance data or the loss function (Menon et al., 2021; Cui et al., 2019) a combination may improve performance further. More work is needed to assess the situations in which the mGEV activation should be used and how much benefit it has in other applications.

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
