# OpenReview forum: "mGEV: Extension of the GEV Activation to Multiclass Classification"
_MIDL.io/2021/Conference/Short — MIDL 2021 Poster_

### Official Review · Reviewer_3Rgx · 2021-04-20

**Confidence:** 4
**Final Rating:** 3

**Summary:**

The paper proposes to extend the GEM activation function from the binary classification setup to the multiclass setup (mGEM) to address the problem of class imbalance. Experiments show that the proposed approach outperforms the softmax baseline on chest x-ray datasets on recall, precision, and F-1 score.

**Strengths:**

1. learning with class imbalance is an important research problem, especially for medical data.
2. The extension from binary to multiclass is simple and intuitive.
3. The proposed approach demonstrates improved performance over the softmax baseline.

**Weaknesses:**

1. The authors should explain why the proposed approach can better address the problem of class imbalance.
2. The paper lacks the baselines of oversampling and loss weighting, which makes it difficult to compare with other approaches.

**Deanonymize Review:**

no

**Detailed Comments:**

In Table 1, not clear why the authors use the weighted average instead of the macro average if the focus is on class imbalance.

**Justification Of The Rating:**

Learning with class imbalance is an important problem and the manuscript demonstrates improved performance over the softmax baseline on the unbalanced x-ray dataset. The authors should elaborate on why the proposed approach is better suited for long-tailed distributions rather than simply presenting the method and results.

**Paper Type:**

methodological development

**Special Issue:**

no

---

### Meta-Review · Program_Chairs · 2021-05-11

**Recommendation:** Accept (Poster)
**Confidence:** 5

**Metareview:**

The authors should supplement more experiments for comparsing with other approaches related to this research problem. However, the proposed method is novel and performance is improved over the existing method, while the authors also release the code to make it is a solid work. In general, I support the opinion of Reviewer 3Rgx and maintain the acceptance for this conference.

---

### Decision · Program_Chairs · 2021-05-11

Accept (Poster)